# DESIGNING LONG-TERM GROUP FAIR POLICIES IN DYNAMICAL SYSTEMS

## ABSTRACT

Neglecting the effect that decisions have on individuals (and thus, on the underlying data distribution) when designing algorithmic decision-making policies may increase inequalities and unfairness in the long term—even if fairness considerations were taken in the policy design process. In this paper, we propose a novel framework for achieving long-term group fairness in dynamical systems, in which current decisions may affect an individual's features in the next step, and thus, future decisions. Specifically, our framework allows us to identify a time-independent policy that converges, if deployed, to the *targeted* fair stationary state of the system in the long-term, independently of the initial data distribution. We model the system dynamics with a time-homogeneous Markov chain and optimize the policy leveraging the Markov chain convergence theorem to ensure unique convergence. We provide examples of different targeted fair states of the system, encompassing a range of long-term goals for society and policy makers. Furthermore, we show how our approach facilitates the evaluation of different long-term targets by examining their impact on the group-conditional population distribution in the long term and how it evolves until convergence.

## 1 INTRODUCTION

The majority of fairness notions that have been developed for trustworthy machine learning (Hardt et al., 2016b; Dwork et al., 2012), assume an unchanging data generation process, i.e., a static system. Consequently, existing work has explored techniques to integrate these fairness considerations into the design of algorithms in static systems (Hardt et al., 2016b; Dwork et al., 2012; Agarwal et al., 2018; Zafar et al., 2017; 2019). However, these approaches neglect the dynamic interplay between algorithmic decisions and the individuals they impact, which have shown to be prevalent in practical settings (Chaney et al., 2018; Fuster et al., 2022). For instance, a decision to deny credit can lead to behavioral changes in individuals as they strive to improve their credit scores for future credit applications. This establishes a feedback loop from decisions to the data generation process, resulting in a shift in the data distribution over time, creating a dynamic system.

Prior research has identified several scenarios where such dynamics can occur, including bureaucratic processes (Liu et al., 2018), social learning (Heidari et al., 2019), recourse (Karimi et al., 2020), and strategic behavior (Hardt et al., 2016a; Perdomo et al., 2020). Existing work on fair decision policies in dynamical systems has examined the effects of policies that aim to maintain existing static group fairness criteria in the short-term, i.e., in two-step scenarios (Liu et al., 2018; Heidari et al., 2019) or over larger amount of time steps (Zhang et al., 2020; Creager et al., 2020; D'Amour et al., 2020). These studies have demonstrated that enforcing static group fairness constraints in dynamical systems can lead to unfair data distributions and may perpetuate or even amplify biases (Zhang et al., 2020; Creager et al., 2020; D'Amour et al., 2020).

Few previous work has attempted to meaningfully extend static fairness notions to dynamic contexts by focusing on the long-term behavior of the system. Existing approaches to learning long-term fair policies (Perdomo et al., 2020; Jabbari et al., 2017; Williams & Kolter, 2019) assume unknown dynamics and learn policies through iterative training within the reinforcement learning framework. While reinforcement learning offers flexibility and is, to some extent, model-agnostic, one of its major drawbacks lies in the requirement for large amounts of training data Henderson et al. (2018); Dulac-Arnold et al. (2021); Wang et al. (2016), alongside the necessity for recurrent policy deployments over time. Successful applications of reinforcement learning typically occur in settings where

a simulator or game is accessible Cutler et al. (2015); Osiński et al. (2020). However, in the real world, we can often not afford to satisfy such requirements.

To address these shortcomings, we propose to separate learning and estimation from decision-making and optimization. We start with a modeling approach of the main relevant (causal) mechanisms of the real world first and require access to a sufficient amount of data to reliably estimate these. The main contribution of this paper then lies in proposing a method of how to use this information to find a policy that leads to a stable long-term fair outcome as an equilibrium state.

We introduce a principle that can be applied to various (causal) models to learn policies aimed at achieving long-term group fairness, along with a computational optimization approach to solve it. Our framework can be thought of as a three-step process: Given sufficient data to estimate (causal) mechanisms, we i) define the characteristics of a long-term fair distribution in the decision-making context; ii) transform this definition into a constrained optimization problem; iii) which we then solve. Importantly, existing long-term group fairness targets (Chi et al., 2022; Wen et al., 2021; Yin et al., 2023; Yu et al., 2022) can be formulated as such long-term fair distribution.

Inspired by previous work (Zhang et al., 2020), we adopt Markov chains as a framework to model system dynamics. We propose an *optimization problem* to find a policy that, if found, guarantees that the system converges, irrespective of the initial state, to the pre-defined targeted fair, stationary data distribution. Such policy offers consistency in decision-making, enhancing stakeholder trust and predictability of decision processes. Furthermore, the policy is guaranteed to converge from any starting distribution, which makes it robust to covariate shift.

Our work differs from research on fair sequential decision learning under feedback loops, where decisions made at one time step influence the training data observed at the subsequent step (Kilbertus et al., 2020b; Rateike et al., 2022a; Bechavod et al., 2019; Joseph et al., 2016). In this scenario, decisions introduce a sampling bias, but do not affect the underlying generative process, as in our case. In our case, decisions influence the underlying data-generating process and consequently shift the data distribution. Our work also diverges from research focused on developing robust machine learning models that can perform well under distribution shifts, where deployment environments may differ from the training data environment (Quinonero-Candela et al., 2008). Unlike the line of research that considers various sources of shift (Makar & D'Amour, 2022; Adragna et al., 2020; Schrouff et al., 2022), our approach leverages policy-induced data shifts to guide the system towards a state that aligns with our defined long-term fairness objectives. Rather than viewing data shifts as obstacles to overcome, we utilize them as a means to achieve fairness goals in the long term.

While our framework can be applied to various dynamical systems, we first provide a guiding example (§ 2). We then provide a framework for policy makers to design fair policies that strategically use system dynamics to achieve effective fair algorithmic decision-making in the long term (§ 3) together with a general optimization problem that allows solving it computationally (§ 5). We then exemplify targeted fair states for the system, leveraging existing fairness criteria (§ 6). Following previous work (Creager et al., 2020; D'Amour et al., 2020), we use simulations to systematically explore the convergence and behavior of different long-term policies found by our framework (§ 7). We conclude with a discussion (§ 8), followed by a summary and outlook (§ 9).

## 2    GUIDING EXAMPLE

We present a guiding example. Note, however, that our framework can also be applied framework to other generative processes (see Appendix F). We assume a data generative model for a credit lending scenario (Liu et al., 2018; Creager et al., 2020; D'Amour et al., 2020) (see Figure 1).

**Data generative model.**    Let an individual with protected attribute $S$ (e.g. gender) at time $t$ be described by a non-sensitive feature $X_t$ (e.g. credit score as a summary of monetary assets and credit history) and an outcome of interest $Y_t$ (e.g. repayment ability). We assume the sensitive attribute to remain immutable over time and drop the attribute's time subscript. For simplicity, we assume binary sensitive attribute and outcome of interest $S, Y \in \{0, 1\}$ and a one-dimensional discrete non-sensitive feature $X \in \mathbb{Z}$. Let the population's sensitive attribute be distributed as $\gamma(s) := \mathbb{P}(S = s)$ and remain constant over time. We assume $X$ to depend on $S$, such that the group-conditional feature distribution at time $t$ is $\mu_t(x \mid s) := \mathbb{P}(X_t = x \mid S = s)$.

For example, different demographic groups may have different credit score distributions due to structural discrimination in society. The outcome of interest is assumed to depend on $X$ and (potentially) on $S$ resulting in the label distribution $\ell(y \mid x, s) := \mathbb{P}(Y_t = y \mid X_t = x, S = s)$. For example, payback probability may be tied to factors like income, which can be assumed to be encompassed within a credit score. We assume that there exists a policy that takes binary loan decisions based on $X$ and (potentially) $S$ and decides with probability $\pi(d \mid x, s) := \mathbb{P}(D_t = d \mid X_t = x, S = s)$. Consider dynamics where a decision $D_t$ at time step $t$ directly influences an individual's features $X_{t+1}$ at the next step. We assume the transition from the current feature state $X_t$ to the next state $X_{t+1}$ depends additionally on the current features, outcome $Y_t$, and (possibly) the sensitive attribute $S$. For example, after a positive lending decision, an individual's credit score may rise due to successful loan repayment, with the extent of in-

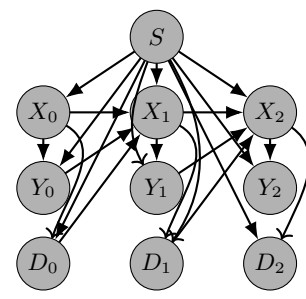

Figure 1: Data generative model. Time steps (subscript) $t = \{0, 1, 2\}$.

crease (potentially) influenced by their sensitive attribute. Let the probability of an individual with $S = s$ transitioning from a credit score of $X_t = x$ to $X_{t+1} = k$ in the next step, denoted as the dynamics $g(k \mid x, d, y, s) := \mathbb{P}(X_{t+1} = k \mid X_t = x, D_t = d, Y_t = y, S = s)$ Importantly, the next step feature state depends only on the present feature state, and not on any past states.

**Dynamical System.** We can now describe the evolution of the group-conditional feature distribution $\mu_t(x \mid s)$ over time $t$. The probability of a feature change from $X_t = x$ to $X_{t+1} = k$ in the next step given $S = s$ is obtained by marginalizing out $D_t$ and $Y_t$, resulting in

$$\mathbb{P}(X_{t+1} = k \mid X_t = x, S = s) = \sum_{d,y} g(k \mid x, d, y, s) \pi(d \mid x, s) \ell(y \mid x, s). \tag{1}$$

These transition probabilities together with the initial distribution over states $\mu_0(x \mid s)$ define the behavior of the dynamical system. In our model, we assume time-independent dynamics $g(k \mid x, d, y, s)$, where feature changes in response to decisions and individual attributes remain constant over time (e.g., through a fixed bureaucratic policy determining credit score changes based on re-payment behavior). We also assume that the distribution of the outcome of interest conditioned on an individual's features $\ell(y \mid x, s)$ remains constant over time (e.g., individuals need certain assets, summarized in a credit score, to repay). Additionally, we assume that the policy $\pi(d \mid x, s)$ can be chosen by a policy maker and may depend on time. Under these assumptions, the probability of a feature change depends solely on policy $\pi$ and sensitive feature $S$.

**Targeted Fair Distribution.** Consider a bank using policy $\pi$ for loan approvals. While maximizing total profit, the bank also strives for fairness by achieving equal credit score distribution across groups (D'Amour et al., 2020). This means, at time $t$ the probability of having a credit score $x$ should be equal for both sensitive groups: $\mu_t(x \mid S = 0) = \mu_t(x \mid S = 1)$ for all $x \in \mathcal{X}$. If credit scores are equally distributed, the policy maker aims to preserve this equal distribution in the next time step:

$$\mu_{t+1}(k \mid s) = \sum_x \mu_t(x \mid s) \mathbb{P}(X_{t+1} = k \mid X_t = x, S = s) \tag{2}$$

for all $k \in \mathcal{X}$, $s \in \{0, 1\}$. This means, the credit score distribution remains unchanged (stationary) when multiplied by the transition probabilities defined above. The policy maker's task is then to find a policy $\pi$ that guarantees the credit score distribution to converge to the targeted fair distribution.

## 3 DESIGNING LONG-TERM FAIR POLICIES

After having introduced the guiding example, we now move to a more general setting of time-homogeneous Markov chains that depend on a policy and sensitive features.

### 3.1 BACKGROUND: TIME-HOMOGENEOUS MARKOV CHAINS

We remind the reader of the formal definition of time-homogeneous Markov chains with discrete states space and draw on the following literature for definitions (Freedman, 2017). For a formulation for general state spaces refer to the Appendix A or (Meyn & Tweedie, 2012b).

**Definition 3.1** (Time-homogeneous Markov Chain). *A time-homogeneous Markov chain on a discrete space $\mathcal{Z}$ with transition probability $P$ is a sequence of random variables $(Z_t)_{t \in T}$ with joint distribution $\mathbb{P}$, such that for every $t \in T$ and $z, w \in \mathcal{Z}$ we have $\mathbb{P}(Z_{t+1} = w \mid Z_t = z) = P(z, w)$.*

In a Markov chain, each event's probability depends solely on the previous state. Recall that the transition probabilities must satisfy $P(z, w) \geq 0$ for all $z, w$, and $\sum_w P(z, w) = 1$ for all $z$. The guiding example can be seen as a Markov chain with state space $\mathcal{X}$ and transition probabilities (1). We have stated that the policy maker aims to achieve a fair stationary distribution (2). To formally define this, we introduce the following concept:

**Definition 3.2** (Stationary Distribution). *A stationary distribution of a time-homogeneous Markov chain $(\mathcal{Z}, P)$ is a probability distribution $\mu$, such that $\mu = \mu P$. More explicitly, for every $w \in \mathcal{Z}$ the following needs to hold: $\mu(w) = \sum_z \mu(z) \cdot P(z, w)$.*

In words, the distribution $\mu$ remains unchanged when multiplied by the transition kernel $P$.

### 3.2 The Objective for Long-term Fair Policies

We generalize the provided example to time-homogeneous Markov chains that depend on a policy $\pi$ and a sensitive attribute $S$. The population's feature distribution over time is represented by a time-homogeneous Markov chain $(Z_t)_{t \in T}$ with a general state space $\mathcal{Z}$. The transition probabilities that depend on the sensitive attribute $S$ and policy $\pi$ are captured by the transition probabilities $P_\pi^s$. Suppose a policy maker aims to achieve a fair distribution $(\mu^s)_{s \in \mathcal{S}}$. The goal for the policy maker is then to find a distribution $(\mu^s)_{s \in \mathcal{S}}$ and policy $\pi$ such that the induced kernel $P_\pi^s$ converges to the distribution $(\mu^s)_{s \in \mathcal{S}}$, and the distribution $(\mu^s)_{s \in \mathcal{S}}$ satisfies the defined fairness constraints.

Now, consider a scenario where our society is already in a fair state $(\mu^s)_{s \in \mathcal{S}}$. In this case, the policy maker would aim to find policy $\pi$ that defines a transition probability $P_\pi^s$ such that the next state remains fair. More formally, we would seek to satisfy the following equation:

$$\mu^s = \mu^s P_\pi^s \tag{3}$$

for all $s \in \mathcal{S}$. This can be seen as a generalization of (2). Therefore, the fair distribution $(\mu^s)_{s \in \mathcal{S}}$ should be the stationary distribution of the Markov chain defined by $(\mathcal{Z}, P_\pi^s)$. Any policy that aims for the fair stationary state $(\mu^s)_{s \in \mathcal{S}}$ will eventually need to find a policy that satisfies (3) to at least transition from a fair state to a fair state in the long term. In this sense (3) defines the fundamental problem of finding long-term fair policies in these settings. To find a policy that ensures convergence to the desired fair distribution, we present a general optimization problem in § 5. This utilizes the Markov Convergence Theorem, which we discuss next.

## 4 Background on Markov Chain Convergence Theorem

The Markov Convergence Theorem establishes conditions for a time-homogeneous Markov chain to converge to a unique stationary distribution, regardless of the initial distribution. In our model, the transition probabilities depend on the sensitive attribute, and we will apply in (4) the Markov Convergence theorem separately to each group's transition probabilities. We thus drop the superscript $s$.

**Theorem 4.1** (Markov Convergence Theorem). *Let $(Z_t)_{t \in T}$ be an irreducible and aperiodic time-homogeneous Markov chain with discrete state space $\mathcal{Z}$ and transition matrix $P$. Then the marginal distribution $\mathbb{P}(Z_t)$ converges to the unique stationary distribution $\mu$ as $t$ approaches infinity (in total variation norm), regardless of the initial distribution $\mathbb{P}(Z_0)$.*

In words, the Markov Convergence Theorem states that, regardless of the initial distribution, the state distribution of an irreducible and aperiodic Markov chain eventually converges to the *unique* stationary distribution. We now provide definitions for irreducibility and aperiodicity.

**Definition 4.2** (Irreducibility). *A time-homogeneous Markov chain is considered irreducible if, for any two states $z, w \in \mathcal{Z}$, there exists a $t > 0$ such that $P^t(z, w) > 0$, where $P^t(z, w) = \mathbb{P}(Z_t = w \mid Z_0 = z)$ represents the probability of going from $z$ to $w$ in $t$ steps.*

In other words, irreducibility ensures that there is a positive probability of reaching any state $w$ from any state $z$ after some finite number of steps. Note, for discrete state space $\mathcal{Z}$, every irreducible time-homogeneous Markov chain has a unique stationary distribution (Thm. 3.3 (Freedman, 2017)).

**Definition 4.3** (Aperiodicity). *Consider an irreducible time-homogeneous Markov chain $(\mathcal{Z}, P)$. Let $R(z) = \{t \geq 1 : P^t(z, z) > 0\}$ be the set of return times from $z \in \mathcal{Z}$, where $P^t(z, z)$ represents the probability of returning to state $z$ after $t$ steps. The Markov chain is aperiodic if and only if the greatest common divisor (gcd) of $R(z)$ is equal to 1: $gcd(R(z)) = 1$ for all $z$ in $\mathcal{Z}$.*

In words, aperiodicity refers to the absence of regular patterns in the sequence of return times to state $z$, i.e., the chain does not exhibit predictable cycles or periodic behavior.

For general state spaces the Markov Convergence Theorem can be proven under Harris recurrence, aperiodicity and the existence of a stationary distribution (Meyn & Tweedie, 2012b) (see Apx. A).

## 5 THE OPTIMIZATION PROBLEM

We now reformulate objective (3) into a computationally solvable optimization problem for finding a time-independent policy. This policy, if deployed, leads the system to convergence to a fair stationary state in the long term, regardless of the initial data distribution.

**Definition 5.1** (General Optimization Problem). *Assume a time-homogeneous Markov chain $(\mathcal{Z}, P_\pi)$ defined by a state space $\mathcal{Z}$ and a kernel $P_\pi^s$. To find policy $\pi$ that ensures the Markov Chain's convergence to a unique stationary distribution $(\mu^s)_{s \in \mathcal{S}}$, while minimizing a fair long-term objective $J_{LT}$ and adhering to a set of fair long-term constraints $C_{LT}$, we propose the following optimization problem:*

$$\min_\pi \quad J_{LT}((\mu^s)_{s \in \mathcal{S}}, \pi) \qquad subj. \ to \quad C_{LT}((\mu^s)_{s \in \mathcal{S}}, \pi) \geq 0; \quad C_{conv}(P_\pi^s) \geq 0 \, \forall s \qquad (4)$$

*where $C_{conv}$ are convergence criteria according to the Markov Convergence Theorem.*

In words, we aim to find a policy $\pi$ that minimizes a long-term objective $J_{\text{LT}}$ subject to long-term constraints $C_{\text{LT}}$ and convergence constraints $C_{\text{conv}}$. The objective $J_{\text{LT}}$ and constraints $C_{\text{LT}}$ are dependent on the policy-induced stationary distribution $(\mu^s)_{s \in \mathcal{S}}$, which represents the long-term equilibrium state of the data distribution and may also depend directly on the policy $\pi$. In § 6, we provide various instantiations of long-term objectives and constraints to illustrate different ways of parameterizing them. Convergence constraints $C_{\text{conv}}$ are placed on the kernel $P_\pi^s$ and guarantee convergence of the chain to a *unique stationary distribution for any starting distribution* according to the Markov Convergence Theorem (Def.4.1). The specific form of $C_{\text{conv}}$ depends on the properties of the Markov chain, such as whether the state space is finite or continuous. In the following, we refer to the notation $\mu_\pi(x \mid s)$ when we are interested in $(\mu^s)_{s \in \mathcal{S}}$ at certain values $x$ and $s$.

**Solving the Optimization Problem.** In our example, the Markov chain is defined over a categorical feature $X$ (credit score), resulting in a finite state space. In this case, the optimization problem becomes a linear constrained optimization problem and we can employ any efficient black-box optimization methods for this class of problems (e.g., Kraft (1988)). We detail this for our example: The convergence constraints $C_{\text{conv}}$ are determined by the aperiodicity and irreducibility properties of the corresponding Markov kernel (see § 4). A sufficient condition for irreducibility is $\texttt{Irred}(\pi) := \sum_{i=1}^n (T_\pi^s)^n \geq \mathbf{0} \, \forall s$, where $n$ is the number of states ($n = |X|$), and $\mathbf{0}$ denotes the matrix with all entries equal to zero. A sufficient condition for aperiodicity requires that the diagonal elements of the Markov kernel are greater than zero: $\texttt{Aperiod}(\pi) := T_\pi^s(x, x) > 0 \, \forall x, s$. The group-dependent stationary distribution $\mu_\pi^s$ based on $T_\pi^s$ can be computed via eigendecomposition (Weber, 2017). In the next section we introduce various objective functions $J_{\text{LT}}$ and constraints $C_{\text{LT}}$ that capture notions of profit, distributional, and predictive fairness. Importantly, for finite state spaces, these objectives and constraints are linear. While our general optimization problem remains applicable in the context of an infinite state space, solving it becomes more challenging due to the potential introduction of non-linearities and non-convexities.

## 6 TARGETED FAIR STATES

Our framework enables users to define their preferred long-term group fairness criteria. Here, we present examples of how long-term fair targets can be quantified by defining a long-term objective $J_{\text{LT}}$ and long-term constraints $C_{\text{LT}}$ in (4). We provide these examples assuming discrete $X$ and binary $D, Y, S$ as in our guiding example (§ 2). Note, our framework allows enforcing common long-term fairness and reward notions (see Appendix B.1).

## 6.1 PROFIT

Assume that when a granted loan is repaid, the bank gains a profit of $(1-c)$; when a granted loan is not repaid, the bank faces a loss of $c$; and when no credit is granted, neither profit nor loss occurs. We quantify this profit as utility (Kilbertus et al., 2020c; Corbett-Davies et al., 2017), considering a cost associated with positive decisions denoted by $c \in [0,1]$, in the following manner: $\mathcal{U}(\pi; c) = \sum_{x,s} \pi(D=1 \mid x,s) \left(\ell(Y=1 \mid x,s) - c\right) \mu_\pi(x \mid s)\gamma(s)$, where $\pi(D=1 \mid x,s)$ is the probability of a positive policy decision, $\ell(y \mid x,s)$ the positive ground truth distribution, $\mu_\pi(x \mid s)$ the stationary group-dependent feature distribution, and $\gamma(s)$ the distribution of the sensitive feature.

A bank's objective may be to maximize utility (minimize financial loss, i.e., $J_{\text{LT}} := -\mathcal{U}(\pi, c)$). In contrast, a non-profit organization may aim to constrain its policy by maintaining a minimum profit level $\epsilon \geq 0$ over the long term to ensure program sustainability ($C_{\text{LT}} := \mathcal{U}(\pi; c) - \epsilon$).

## 6.2 DISTRIBUTIONAL FAIRNESS

Policy makers may also be interested in specific characteristics of a population's features $X$ or qualifications $Y$ (ground truth) on a group level. We measure group qualification $\mathcal{Q}$ as the group-conditioned proportion of positive labels assigned to individuals (Zhang et al., 2020) as $\mathcal{Q}^s(\pi \mid s) = \sum_x \ell(Y=1 \mid x,s)\mu_\pi(x \mid s)$, where $\ell(Y=1 \mid x,s)$ is the positive ground truth distribution, and $\mu_\pi(x \mid s)$ describes the stationary group-dependent feature distribution. We measure inequity (of qualifications) as $\mathcal{I} := \mid \mathcal{Q}(\pi \mid S=0) - \mathcal{Q}(\pi \mid S=1) \mid$.

To promote financial stability, a policy maker like the government may pursue two different objectives. Firstly, they may aim to minimize default rates using the objective $J_{\text{LT}} := -\sum_s \mathcal{Q}(\pi \mid s)\gamma(s)$. Alternatively, if the policy maker intends to increase credit opportunities, they may seek to maximize the population's average credit score with the objective $J_{\text{LT}} := -\sum_s \frac{1}{|X|} \sum_x \mu_\pi(x \mid s)\gamma(s)$, where $|X|$ represents the state space size. To achieve more equitable credit score distributions, the policy maker could impose the constraint $C_{\text{LT}} := \epsilon - \mid \mu_\pi(x \mid S=0) - \mu_\pi(x \mid S=1) \mid \forall x$. However, depending on the generative model, this approach might not eliminate inequality in repayment probabilities. In such cases, the policy maker may aim to ensure that individuals have the same payback ability using the constraint $C_{\text{LT}} := \epsilon - \mathcal{I}$. Note that measuring differences in continuous or high-dimensional distributions requires advanced distance measures. Additionally, prioritizing egalitarian distributions may not always align with societal preferences (Barsotti & Koçer, 2022; Martinez et al., 2020) (see Appendix C). Finally, equal credit score distributions or repayment probabilities may not guarantee equal access to credit, we thus next introduce predictive group fairness measures.

## 6.3 PREDICTIVE FAIRNESS

Ensuring long-term predictive fairness can help a policy maker meet regulatory requirements and maintain public trust. One example of a predictive group unfairness measure is *equal opportunity* (Hardt et al., 2016b): $\text{EOPUnf}(\pi) = |\mathbb{P}_\pi(D=1 \mid Y=1, S=0) - \mathbb{P}_\pi(D=1 \mid Y=1, S=1)|$. This measures the disparity in the chance of loan approval for eligible loan applicants based on their demographic characteristics. Note: $\mathbb{P}_\pi(D=1 \mid Y=1, S=s) = \frac{\sum_x \pi(D=1 \mid x,s)\ell(Y=1 \mid x,s)\mu_\pi(x \mid s)}{\sum_x \ell(Y=1 \mid x,s)\mu_\pi(x \mid s)}$.

In the fairness literature, it is common for a policy maker to define a maximum tolerable unfairness threshold as $\epsilon \geq 0$, expressed as $C_{\text{LT}} := \epsilon - \text{EOPUnf}$. Alternatively, they may aim to minimize predictive unfairness $\text{EOPUnf}$ over the long term by imposing $J_{\text{LT}} := \text{EOPUnf}(\pi)$. Note, our framework also allows for other group fairness criteria, such as demographic parity (Dwork et al., 2012) or sufficiency (Chouldechova, 2017).

In this section, we presented various long-term goals as illustrative examples for lending policies. For methods to impose constraints on the types of policies under consideration, please refer to Appendix C. This section serves as a starting point for discussions on these objectives and we encourage the exploration of a wider range of long-term targets by drawing inspiration from existing research in social sciences and economics, while also involving affected communities in defining these objectives. In the following section, we demonstrate how our approach enhances the understanding of the interplay between diverse long-term goals and constraints.

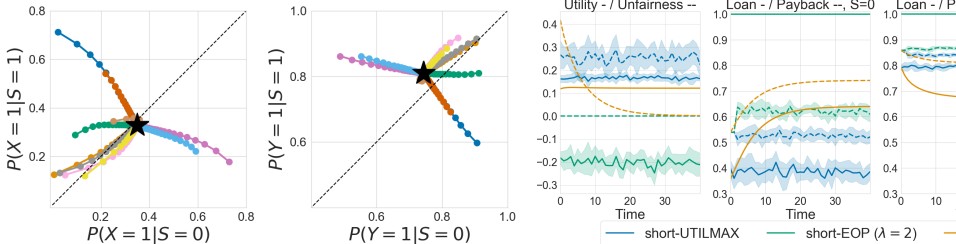

(a) Convergence: $\pi^\star_{\text{EOP}}$ to unique stationary distribution $\star$. 200 time steps. Colors: 10 random initial feature distributions. Feature $X = 1$ left, outcome $Y$ right. Equal distribution dashed.

(b) Utility (solid, $\uparrow$), EOP-Unfairness (dashed, $\downarrow$) for short-term-UTILMAX (unfair), short-term-EOP policies (10 seeds), our long-term-EOP policy. Loan (solid) and payback probab. (dashed) per sensitive $S$.

Figure 2: (a) Convergence independent of initial distribution. (b) Comparison to short-term policies.

## 7 SIMULATIONS

We validate our proposed optimization problem formulation in semi-synthetic simulations. Using our guiding example with real-world data and assumed dynamics, we first demonstrate that the policy solution, if found, converges to the targeted stationary state (§ 7.1). Then, we demonstrate how our approach helps to analyze the interplay between long-term targets and dynamics (§ 7.2). For additional results see Appendix E. Our code is available at `github.com/XXXX`.

**Data and General Procedure.** We use the real-world FICO loan repayment dataset (Reserve, U. F., 2007), with data pre-processing from (Barocas et al., 2019). It includes a one-dimensional credit score $X$, which we discretize into four bins for simplicity, and a sensitive attribute $S$ that we binarize: Caucasian ($S = 1$) and African American ($S = 0$). From this dataset, we estimate the initial feature distribution $\mu_0(x \mid s)$, label distributions $\ell(y \mid x, s)$, and sensitive group ratios $\gamma(s)$. Note, the FICO dataset provides probability estimates. For results under estimated probabilities and dynamics when labels are partially observed, refer to the Appendix E.6. Since FICO is a static dataset, we assume dynamics $g(k \mid x, d, y, s)$. We first apply the general principle (4) to formulate an optimization problem via long-term objectives $J_{\text{LT}}$ and long-term constraints $C_{\text{LT}}$ and convergence constraints $C_{\text{conv}}$. Next, we solve the optimization problem. Using the found policy $\pi^\star$ and the resulting Markov kernel $T_{\pi^\star}$, we generate the feature distribution across 200 steps. See Appendix D for details.

We solve the problem using the Sequential Least Squares Programming method from scikit-learn (Pedregosa et al., 2011), initializing it (warm start) with a uniform policy where all decisions are random ($\pi(D = 1 \mid x, s) = 0.5\ \forall s, x$). See Appendix D for details.

### 7.1 CONVERGENCE TO TARGETED DISTRIBUTION AND TEMPORAL STABILITY

We demonstrate that a policy derived from an optimization problem based on the general principle converges to a stable steady-state distribution. For setup details see Appendix D.

**One-sided Dynamics.** One-sided dynamics are characterized by a particular (usually positive) decision leading to changes in a feature distribution, while other decisions do not incur any feature changes. Following prior work (Liu et al., 2018; D'Amour et al., 2020), we assume in our scenario, that if an applicant defaults on their loan, their credit score remains the same; if the applicant repays the loan, their credit score is likely to increase. We refer to these dynamics as `one-sided`.

**Maximum Utility under EOP-Fairness.** We now exemplify a long-term target. Consider a bank that aims to maximize its profit ($\mathcal{U}$) while guaranteeing equal opportunity (`EOPUnf`) for loan approval. Given cost of a positive decision $c$ and a small unfairness level $\epsilon$, we seek for a policy:

$$\pi^\star_{\text{EOP}} := \arg_\pi \max \mathcal{U}(\pi; c) \qquad \text{subj. to} \quad \text{EOPUnf}(\pi) \leq \epsilon;\ C_{\text{conv}}(T_\pi), \tag{5}$$

This target has been proposed for fair algorithmic decision-making in static systems (Hardt et al., 2016b), short-term policies aiming to fulfill this target at each time step have examined in dynamical systems (Zhang et al., 2020; Creager et al., 2020; D'Amour et al., 2020) and has been imposed as

long-term target Wen et al. (2021). We redefine this concept as a long-term goal for the stationary distribution to satisfy.

**Results.** We run simulations on 10 randomly sampled initial feature distributions $\mu_0(x \mid s)$, setting $\epsilon = 0.01, c = 0.8$. Figure 2a displays the resulting trajectories of the feature distribution for $X_1$ converging to a stationary distribution. For other features see Appendix E.1). We observe that while the initial distribution impacts convergence process and time, the policy consistently converges to a single stationary distribution regardless of starting point. The policy found for one population can thus be effectively applied to other populations with different feature distributions, if dynamics and labeling distributions remain unchanged. As the outcome of interest $Y$ depends on the features, its distribution converges also to a stationary point.

We now compare our found long-term fair policy to both fair and unfair short-term policies. Figure 2b displays $\mathcal{U}$ and EOPUnf. Using the initial distribution $\mu_0(x \mid s)$ from FICO, we solve the optimization problem (5) for tolerated unfairness $\epsilon = 0.026$. The short-term policies consist of Logistic Regression models for 10 random seeds, which are retrained at each time step; fairness is enforced using a Lagrangian approach ($\lambda = 2$). Our policy demonstrates high stability in both utility and fairness compared to short-term policies, which exhibit high variance across time. Note since our policy does not require training, we do not report standard deviation over different seeds. Furthermore, while our policy converges to the same fairness level as the short-term fair policy, it experiences only a marginal reduction in utility compared to the (unfair) utility-maximizing short-term policy. Thus, it does not suffer from a fairness-utility trade-off to the extent observed in the short-term policies.

Figure 2b (middle, right) displays loan $\mathbb{P}(D=1 \mid S=s)$ and payback probabilities $\mathbb{P}(Y=1 \mid S=s)$ for non-privileged ($S = 0$) and privileged ($S = 1$) groups. The short-term fair policy achieves fairness by granting loans to everyone. For the utility-maximizing short-term policy, unfairness arises as gap between ability to pay back and loan provision is much smaller for the privileged group, resulting in significantly different loan probabilities between the two groups. For our long-term policy, we observe that loan provision probabilities converge closely for both groups over time, while the gap between payback probability and loan granting probability remains similar between groups. Similar to prior research (Wen et al., 2021; Yu et al., 2022), we observe that our policy achieves long-term objectives, but the convergence phase may pose short-term fairness challenges. In practice, it is essential to assess the potential impact of this on public trust.

## 7.2 LONG-TERM EFFECTS OF TARGETED STATES

This section examines the long-term effects of policies and their targeted stationary distributions. The observations are specific to the assumed dynamics and distributions and serve as a starting point for a thoughtful reflection on the formulation and evaluation of long-term targets.

**Maximum Qualifications.** Inspired by (Zhang et al., 2020), assume a non-profit organization offering loans. Their goal is to optimize the overall payback ability ($\mathcal{Q}$) of the population to promote societal well-being. Additionally, they aim to sustain their lending program by prevent non-negative profits ($\mathcal{U}$) in the long-term. We thus seek for:

$$\pi_{\text{QUAL}}^{\star} := \arg_\pi \max \mathcal{Q}(\pi) \qquad \text{subj. to} \quad \mathcal{U}(\pi) \geq 0; \; C_{\text{conv}}(T_\pi) \qquad (6)$$

**Two-sided Dynamics.** In addition to one-sided dynamics, where only positive decisions impact the future, we also consider two-sided dynamics (Zhang et al., 2020), where both positive and negative decisions lead to feature changes. We investigate two types of two-sided dynamics. Under recourse dynamics, individuals receiving unfavorable lending decisions take actions to improve their credit scores, facilitated through recourse (Karimi et al., 2021b) or social learning (Heidari et al., 2019). In discouraged dynamics, unfavorable lending decisions demotivate individuals, causing a decline in their credit scores. This may happen when individuals cannot access loans for necessary education, limiting their financial opportunities.

**Results.** We solve both introduced optimization for policies $\pi_{\text{EOP}}^{\star}$(5) and $\pi_{\text{QUAL}}^{\star}$(6) with $c = 0.8$ and $\epsilon = 0.01$, both subject to convergence constraints $C_{\text{conv}}$ (irreducibility, aperiodicity), for one-sided, recourse and discouraged dynamics. Utilizing the found policies we simulate the feature distribution over 200 time steps, starting from

the initial FICO feature distribution. For more details, refer to Appendix D. Figure 3 shows accumulated (effective) measures of utility, inequity and EOP-Unfairness over time.

Across different dynamics, the policies conform with their targets. $\pi^\star_{\text{EOP}}$ accumulates across dynamics most utility, while $\pi^\star_{\text{QUAL}}$ has a small negative cumulative utility due to the imposed zero-utility constraint. In the `one-sided` scenario, we observe for unfairness different short-term and long-term effects. Up to approx. 40 time steps, $\pi^\star_{\text{QUAL}}$ yields lower unfairness than $\pi^\star_{\text{EOP}}$, after this point $\pi^\star_{\text{QUAL}}$ becomes highly unfair. These observations highlight that: dynamics may significantly impact the final outcome of decision policies; when deploying a policy in the long-term small differences in policies can lead to large accumulated effects; and short term effects may differ from long-term goals.

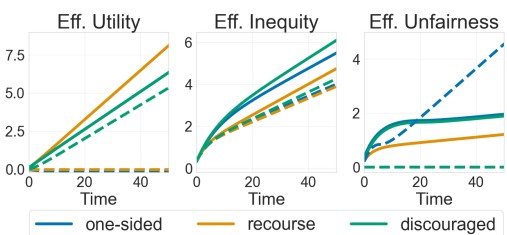

Figure 3: Effective utility $\mathcal{U}$, inequity $\mathcal{I}$ and `EOPUnf` for policies $\pi^\star_{\text{EOP}}$ (solid), $\pi^\star_{\text{QUAL}}$ (dashed) and different dynamics: `one-sided`, and two-sided: `recourse`, `discouraged`.

## 8 DISCUSSION

In this section, we discuss key assumptions and limitations. Additional discussion in Appendix B.

**Limitations of Assumptions.** The proposed general optimization problem (4) assumes a time-homogeneous kernel and access to the dynamics defining it. Although real-world data often change over time, we treat the dynamics as static for a shorter duration, which is plausible, if they rely on bureaucratic (Liu et al., 2018) or algorithmic recourse policies (Karimi et al., 2022), and if convergence time remains relatively short, as seen in our simulations. However, convergence time depends on the dynamics and initial distribution. If the transition probabilities become time-dependent, updating the policy would be necessary. Transition probabilities for discrete state spaces can be estimated from temporal data (Sherlaw-Johnson et al., 1995; Craig & Sendi, 2002), but remains a challenge for continuous state spaces in practice (Duffie & Glynn, 2004). Furthermore, few temporal datasets for fair machine learning exist (Mehrabi et al., 2019). Assuming dynamics with expert knowledge is an alternative, but caution is needed as it may lead to confirmation bias (Nickerson, 1998).

**The Case of Non-existence of a Long-Term Fair Policy.** Consider the case that no solution exists for our problem (3). Then, as argued in § 3, no policy maker with different strategies of finding policies over time would find a solution to the same problem, with the same assumed distributions, dynamics, and constraints. If a solution to our optimization problem does not exist, this insight may prompt practitioners to explore alternative approaches for long-term fairness, such as non-stationary objectives (Zhang et al., 2020) or redefining the fair state. Thus, our approach enhances the understanding of system dynamics and long-term fairness.

## 9 SUMMARY AND OUTLOOK

We have introduced a general problem for achieving long-term fairness in dynamical systems, where algorithmic decisions in one time step impact individuals' features in the next time step, which are consequently used to make decisions. We proposed an optimization problem for identifying a time-independent policy that is guaranteed to converge to a targeted fair stationary state, regardless of the initial data distribution. We model the system dynamics with a time-homogeneous Markov chain and enforce the conditions of the Markov chain convergence theorem to the Markov kernel through policy optimization. Our framework can be applied to different dynamics and long-term fair goals as we have shown in a guiding example on credit lending. In semi-synthetic simulations, we have shown the effectiveness of policy solutions to converge to targeted stationary population states in a stable manner. Future work lies in applying our framework to a wider range of problems with more complex dynamics, and larger (potentially continuous) feature spaces. Future work may also explore the application of our framework to designing social interventions on the transition probabilities (Heidari et al., 2019; von Kügelgen et al., 2022; Mhasawade & Chunara, 2021) providing additional insights and solutions for long-term fairness in algorithmic decision-making in dynamical systems.

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
