# OpenReview forum: "Designing Long-term Group Fair Policies in Dynamical Systems"
_ICLR.cc/2024/Conference — Submitted to ICLR 2024_

### Official Review · Reviewer_NkS5 · 2023-10-31

**Soundness:** 2 fair
**Presentation:** 2 fair
**Contribution:** 1 poor
**Rating:** 3
**Confidence:** 4

**Summary:**

This paper proposes a general framework using Markov Chains to model long-term fairness in dynamic system where decisions can change the underlying states. Modeling the problem as a constrained optimization, a time independent policy can be identified using Markov Chain convergence theorem. Examples of long-terms fairness formulation were given together with simulations.

**Strengths:**

1.	The model has a general formulation which can incorporate different long-term fairness objectives.
2.	The convergence to the stationary distribution is guaranteed as long as the assumptions for Markov Convergence Theorem are satisfied.
3.	The paper is well organized with good background summary.

**Weaknesses:**

1.	The main weakness of the paper is in the contribution. As claimed in the introduction, the drawback of using RL based methods for unknown dynamics is the requirement of large amount of training data. In this paper, however, it is assumed that the model is already learned. So the problem is changed to an optimization problem with known parameters. Therefore, it does not seem to be a fair comparison that this is an improvement over RL based methods.
2.	As cited in section 6, there are many existing works with different long-term fairness constraints. It is not clear what is the advantage of having a unified framework. For example, can this new framework solve the problem faster especially for more complex dynamics with large state space? Otherwise, the new model framework seems like a reformulation of existing models.

**Questions:**

1.	The key operation of the framework is solving the optimization problem as described in section 5. It would be better to give more information on the details of the complexity of solving this constrained optimization problem.
2.	The description of the simulation is not very clear. For example, from the results, what is the target fairness state? Why would $P(Y=1|S=0)$ be different from $P(Y=1|S=1)$?

---

> ### Author Response · Authors · 2023-11-21
>
> Dear Reviewer NkS5,
>
> Firstly, we would like to comment on the differences between our approach and reinforcement learning (RL), similar to our comment on Reviewer LjFL. While there are shared elements with RL, our methodology differs significantly in key aspects. As RL, our policy (agent) makes decisions within an environment, where the environment responds to these decisions (actions). However, in RL, the response of the environment is quantified by feedback in the form of rewards that accumulate over time and are used to iteratively retrain the policy during deployment. Instead, our approach involves solving an optimization problem (OP) to determine a single long-term policy prior to deployment that we keep fixed during deployment. Notably, we do not retrain the policy based on feedback from its actions. Instead, in solving the optimization problem before deployment, we leverage the Markov Convergence properties and incorporate the expected feedback from the environment to determine the optimal policy. Therefore our policy does not aim to optimize for cumulative reward over time as in RL; rather, it focuses solely on the reward obtained at the stationary distribution. This distinction sets our approach apart from traditional RL methodologies.
>
> Second, we believe our approach complements prior work on RL. As opposed to prior work (e.g., in RL), our paper takes a structured approach by separating the estimation problem (of the Markov kernel i.e., the dynamics) from the policy learning process. We believe that this is a strength of our approach, as it allows us to find a single time-invariant policy. In the Appendix we show results, where we estimate label distribution and dynamics from temporal data, when labels are only partially observed (as common in the lending scenario), and solve our OP. For these experiments, we used (comparably to RL) a small number of samples to estimate the distributions. We compare our results of policies based on estimated dynamics and true dynamics. We do recognize that the dynamics estimation problem itself is a significant challenge and requires careful attention and, as commented in Section 8, is the subject of a different line of active research and thus outside the scope of this paper.
>
> We see our main contribution in providing one method for finding long-term fair policies that differ from previously proposed RL methods in the aspects detailed above. Our policy, if found and deployed, guarantees convergence to the fair stationary state. We appreciate your expression of concern that our framework is a reformulation of existing models and would like to ask you to provide us with references that provide a similar contribution.
>
> Our simulations - similar to the ones from prior work [1, 2, 3, 4, 5] - provide a straightforward example as a proof of concept. We acknowledge that the application of our framework to address problems involving complex dynamics and large state spaces is beyond the current scope of our paper. Nevertheless, we recognize this as a crucial avenue for future research.
>
> Your second question is not entirely clear to us. Could you please provide additional clarification? We welcome suggestions on improving the description of the simulation. One of the policies we address is an EOP-fair utility maximization policy. In this context, we aim to find a policy that converges to a fair stationary state that is EOP-fair. From the set of policies converging to an EOP-fair state, our objective is to identify the one maximizing utility. The results presented in Figure 2(b) demonstrate that our long-EOP policy indeed converges to a fair state, with EOP close to zero, and its utility is comparable to that of a short-term unfair policy solely focused on maximizing utility. This indicates that our policy is effectively converging to the targeted fair state.
> We hope that our responses have adequately addressed your questions. We appreciate you raising your score, if our response has alleviated your concerns. Kindly let us know if there is anything else we can clarify.
>
> Best regards
>
> [1] Zhang et al. How do fair decisions fare in long-term qualification?
>
> [2] D’Amour et al. Fairness is not static: deeper understanding of long term fairness via simulation studies.
>
> [3]  Liu et al. Delayed impact of fair machine learning.
> [4] Creager et al. Causal modeling for fairness in dynamical systems.
>
> [5] Chi et al. Towards Return Parity in Markov Decision Processes.

---

### Official Review · Reviewer_nac1 · 2023-11-01

**Soundness:** 2 fair
**Presentation:** 3 good
**Contribution:** 2 fair
**Rating:** 5
**Confidence:** 2

**Summary:**

This paper presents a framework for achieving long-term group fairness in dynamical systems, in which current decisions may affect an individual’s features in the next step, and thus, future decisions. The framework is largely characterized by two steps: defining the characteristics of a long-term fair distribution in the decision-making context, and transforming it into a constrained optimization problem. The framework is evaluated on a loan repayment dataset.

**Strengths:**

1. The paper is clearly presented and easy to follow.
2. The considered problem is interesting and relevant to ICLR.

**Weaknesses:**

First, I must admit that I am not an expert in this field and am unfamiliar with the literature. However, I have some concerns regarding the novelty and technical soundness of this paper.

1. In terms of novelty, this paper appears to be closely related to (Zhang et al., 2020), as acknowledged in the paper. However, the differences between the two have not been adequately discussed. From my perspective, both the models and techniques applied in this paper seem quite similar to those in (Zhang et al., 2020).

2. Additionally, this paper only considers one dataset, which raises questions about the generalizability of the framework to other datasets. For instance, (Zhang et al., 2020) considered two real-world datasets: the FICO dataset (same as in this paper) and the COMPAS dataset (which is not addressed in this paper). It would be much more beneficial to explore the performance of the model on multiple datasets to assess its applicability more comprehensively.

**Questions:**

1. What is the conceptual and technical novelty of this paper, compared to (Zhang et al. 2020)?
2. What is the reason for considering only one dataset?

---

> ### Author Response · Authors · 2023-11-21
>
> Dear Reviewer nac1,
>
> Thank you for your review. We are happy to provide you with additional clarifications on your raised questions.
>
> Question 1) We are happy to elaborate on the conceptual and technical novelty of this paper, compared to (Zhang et al. 2020). There are several differences, mainly i) in the modeling assumptions, ii) in the goal of the paper, and iii) in the results shown.
>
> Firstly, there is a disparity in their assumptions regarding the generative approach, and this divergence significantly affects their analytical capabilities. Zhang et al. (2020) operate under the assumption of a Markov Kernel over a binary target variable $Y$. Their model posits an anti-causal relationship, where the target variable $Y$ influences the features $X$ ($Y \to X$). In contrast, our approach assumes a Markov Kernel over a categorical feature space $X$ and we assume a causal relationship where features $X$ influence the target variable $Y$ ($X \to Y$).
>
> We base our modeling assumptions on several considerations. First, when working with the FICO loans dataset, we argue that it is more plausible for features like income and financial assets, which are aggregated into a risk score $X$, to be the causal factors influencing an individual's repayment behavior $Y. This is in contrast to the notion that individuals possess hidden information about their ability to repay $Y$, causing them to have a specific risk score $X$ that summarizes their financial information. This directional causal modeling aligns with prior research [1, 2, 3]. Secondly, we posit that it is more plausible for feedback to induce changes in the feature $X$ rather than altering the ability to repay $Y$, thus we assume that the Markov Chain is defined over $X$.
>
> To the best of our knowledge, the analysis conducted by Zhang et al. (2020) is not easily adaptable to a causal data generative scenario $X \to Y$ with a Markov Chain defined over $X$. If one were to examine short-term policy updates for a Markov kernel defined over $X$ while assuming a causal relationship $X \to Y$, this would result in a time-varying Markov kernel. This introduces significant challenges in assessing long-term effects, as the conventional Markov Convergence theorem is formulated for time-invariant Markov Chains.
>
> Second, the goals of our papers are fundamentally different. Zhang et al. (2020) focus on assessing the influence of fairness constraints on the disparity of qualification rates within the context of short-term policies. Instead, we propose a framework to learn a policy a long-term fair policy that takes the underlying dynamics into account. The goal of our paper is thus fundamentally different.
>
> Third, Zhang et al. (2020) do have a short section 6, where they explore alternative interventions that can effectively improve qualification rates at the equilibrium and promote equality across different groups. This section appears closest to us. Zhang et al. (2020) show that sub-optimal short-term fair policies instead of the optimal ones can improve overall qualification in the long run, that under certain conditions on transitions, there always exist threshold policies leading to equitable equilibriums, and that instead of performing a policy intervention, one could also alter the value of transitions.
>
> We propose a framework to explicitly learn a policy long-term fair policy that induces a time-invariant Markov Kernel and consequently imposes the necessary conditions from the Markov Convergence theorem on the policy.
>
> Question 2) Regarding the choice of the dataset, we acknowledge that our current experiments focus on a single simulation setup, specifically centered around loan repayment. While conducting experiments across a broader range of environments may be interesting, focusing on a single generative model like Zhang et al. (2020) and a single guiding example is in line with prior published work [2, 3, 4] with the loan example used widely by previous work on long-term fairness [1, 2, 3, 4, 5]. Similarly, in our experiments, we vary dynamics and initial distributions, essentially simulating different datasets of the same generative model. Note also that we provide an example of how the framework can be applied to a different generative model in the Appendix.
> We hope that our responses have adequately addressed your questions. We appreciate you raising your score, if our response has alleviated your concerns. Kindly let us know if there is anything else we can clarify.
>
> Best regards
>
> [1] D’Amour et al. Fairness is not static.
>
> [2]  Liu et al. Delayed impact of fair machine learning.
>
> [3] Creager et al. Causal modeling for fairness in dynamical systems.
>
> [4] Wen et al. Algorithms for fairness in sequential decision making.
>
> [5] Yu et al. Policy Optimization with Advantage Regularization for Long-Term Fairness in Decision Systems.

---

### Official Review · Reviewer_ftJq · 2023-11-02

**Soundness:** 2 fair
**Presentation:** 3 good
**Contribution:** 2 fair
**Rating:** 3
**Confidence:** 4

**Summary:**

The paper examines setting where akin to performative prediction point of view machine learning algorithms via their decisions can affect the data/individuals they operate on creating a closed loop between predictions and datasets. In this type of setting they explore how long-term fairness can be achieved from a dynamical systems perspective. Particularly, the assume that the interaction between the algs and people can be modeled via a Markov chain and under assumptions such as irreducibility, aperiodicity and homogeneity it is guaranteed to converge to a single stationary distribution regardless of initial conditions. Given this setting they define an optimization problem to find a policy such that, if found, guarantees that the system converges, irrespective of the initial state, to a pre-defined targeted fair, stationary
data distribution. In their setting the probem in question is a linear constrained optimization problem and they can readily employ any efficient black-box optimization methods for this class. They consider a handful of different fairness metrics (that can be plugged in in the framework) and end with simultations.

**Strengths:**

The paper examines a very interesting and important problem working on the interplay between ML algorithms and the data they operate on. The framework is flexible and as long as the statespace is not too large allows for solving the problem in question using a plethora of readily available techniques.

**Weaknesses:**

I am not perfectly convinced about the modelling assumptions in the paper. Markovian assumption (memoryless behavior, linear systems) and even further the stronger assumption than the controller has the power to induce a MC with a unique attracting stationary distribution does not seem realistic to me.

First of all, there are recent paper that examine dynamics in such settings but the dynamics driven by the agents strategically adapting their data are non-linear as the dynamics typically are in game theory. Of particular interest in this case is [1], where it is shown that such dynamics can be formally chaotic and in fact have periodic orbits of all possible periods. What this means is that these systems have infinitely many stationary distributions. (take the uniform probability distribution on each periodic orbit of length N for arbitrary N). This also happens for relatively simple settings of performative prediction with standard optimization/game theory dynamics. This is a dynamical system perspective that it is in stark contrast with the current one.

Now there are some models of agent dynamics in the literature using MC (e.g. [3] and follows-ups) but in those papers there is a very expansive description of why the specific choices appropriate. In the current paper, I felt that these assumptions are largely there to make the setting tractable and reducible to a problem that we already know how to solve. Is there any way to connect game theoretic models (e.g. [3]) to yours?

Given that there no new computational or analytic tools developed I believed the main contribution of the paper is on the modeling side, hence I would like to see a more careful discussion of these choices instead of generic theory of MCs.

[1] Piliouras, Georgios, and Fang-Yi Yu. "Multi-agent performative prediction: From global stability and optimality to chaos." Proceedings of the 24th ACM Conference on Economics and Computation. 2023.
[2] Narang, Adhyyan, et al. "Multiplayer performative prediction: Learning in decision-dependent games." Journal of Machine Learning Research 24.202 (2023): 1-56.
[3] Omidshafiei, Shayegan, et al. "α-rank: Multi-agent evaluation by evolution." Scientific reports 9.1 (2019): 9937.

**Questions:**

Why is a linear, memoryless model with appropriate? Why is it a reasonable assumption that the system controller can enforce the assumptions of the Markov Convergence Theorem and instead not have many possible stationary distributions?

---

> ### Author Response · Authors · 2023-11-21
>
> Dear Reviewer ftJq,
>
> Thank you for your review. We are happy to provide you with additional clarifications on your raised questions.
>
> Our intention in developing a framework for long-term fair policy learning is to provide a versatile approach that could be applied across various contexts. While models serve as simplified representations of complex systems, they allow us to analyze phenomena otherwise incomprehensible. Our choice of utilizing Markov Chains as a modeling tool is a reflection of this principle. Markov Chains are chosen for their wide application in understanding dynamic processes. For example, the field of Reinforcement Learning (RL) relies on Markov Decision Processes (MDPs), a specific kind of Markov Chain. The proposed modeling framework can indeed be adapted to a variety of different scenarios and we provide an example of a different scenario / generative model in the Appendix.
>
> Differing from game theory, which inherently focuses on microeconomic concepts and thus individuals with distinct preference functions, our approach takes a macro perspective, considering population distributions. In this, we rely on the modeling assumptions and dynamics established by previously published works to explore long-term scenarios [1, 2, 3]. We detail in the illustrative example, the motivation for modeling assumptions as well as in the appendix.
>
> Our modeling assumptions align with an established line of research, and our contribution complements this prior work. While we acknowledge the interest in examining more complex dynamics, drawing insights from game theory for future exploration, we assert that our contribution should be considered on par with, and not unfairly compared to, the existing body of prior work. To the best of our knowledge, our paper is the first to connect the existing work on the Markov Convergence Theorem to the goal of learning long-term fair policies. Specifically, we propose to impose necessary convergence criteria from the Markov Convergence Theorem and find a long-term fair policy by solving a constrained OP prior to deployment of the policy.
>
> Nevertheless, we appreciate you recommending a more detailed discussion of our modeling choices and we will ensure to incorporate this in the revised version of the paper.
>
> We hope that our responses have adequately addressed your questions. We appreciate you raising your score, if our response has alleviated your concerns. Kindly let us know if there is anything else we can clarify.
>
> Best regards
>
> [1] Alexander D’Amour, Hansa Srinivasan, James Atwood, Pallavi Baljekar, D Sculley, and Yoni Halpern. Fairness is not static: deeper understanding of long term fairness via simulation studies. In Proceedings of the 2020 Conference on Fairness, Accountability, and Transparency, pages 525–534, 2020.
>
> [2] Lydia T Liu, Sarah Dean, Esther Rolf, Max Simchowitz, and Moritz Hardt. Delayed impact of fair machine learning. In Proceedings of the 35th International Conference on Machine Learning, ICML 2018, 2018.
>
> [3] Elliot Creager, David Madras, Toniann Pitassi, and Richard Zemel. Causal modeling for fairness in dynamical systems. In International Conference on Machine Learning, pages 2185–2195. PMLR, 2020.

---

> > ### Comment · Reviewer_ftJq · 2023-11-22
> > **Response**
> >
> > Although our appreciate your response I do not feel that my questions have been satisfactorily addressed. In my reasoning above, I have raised some specific concerns why I find these choices to not be suitable. I would be much more convinced with an answer that pointed out which part of my reasoning was incorrect that mere pointers to prior work.
> >
> > Also, I am still not certain in what way your modeling assumptions are justified by [1,2,3]. E.g. a cursory look though [2] reveals that the term Markov is never used in the paper (let along the stronger assumption of a unique stationary distribution). Maybe I am missing something but at the moment I do not see any direct connection between the two models.

---

### Official Review · Reviewer_LjFL · 2023-11-05

**Soundness:** 3 good
**Presentation:** 3 good
**Contribution:** 2 fair
**Rating:** 3
**Confidence:** 3

**Summary:**

This paper considers fair RL and proposes an algorithm that converges to a desirable fair policy. The paper also discussed several fairness measurements and conduct extensive simulations.

**Strengths:**

The paper is well-written. The fairness issue is important for RL. The discussions on fairness are clear. The simulations are extensive.

**Weaknesses:**

The paper lacks enough novelty to be accepted by ICLR. There is a significant amount of space used to explain basic MDP and Markov Chain properties. The novel contribution of this paper is rather limited. Theorem 4.1 is an established result instead of the authors' new contributions. The discussions on different definitions of fairness are also standard. In summary, I struggle to see the novelty of this paper.

**Questions:**

Q: what's the novelty of the proposed method? What's the technical challenges behind it?

---

> ### Author Response · Authors · 2023-11-21
>
> Dear Reviewer LjFL,
>
> Thank you for your review. We are happy to provide you with additional clarifications on your raised questions.
>
> Firstly, we would like to clarify that we believe our approach should not be categorized within the realm of reinforcement learning (RL), contrary to your indication in the summary. While there are shared elements with RL, our methodology differs significantly in key aspects. As RL, our policy (agent) makes decisions within an environment, where the environment responds to these decisions (actions).
>
> However, in RL, the response of the environment is quantified by feedback in the form of rewards that accumulate over time and is used to iteratively retrain the policy during deployment. Instead, our approach involves solving an optimization problem (OP) to determine a single long-term policy prior to deployment that we keep fixed during deployment. Notably, we do not retrain the policy based on feedback from its actions. Instead, in solving the optimization problem before deployment, we leverage the Markov Convergence properties and incorporate the expected feedback from the environment to determine the optimal policy. Therefore our policy does not aim to optimize for cumulative reward over time as in RL; rather, it focuses solely on the reward obtained at the stationary distribution. This distinction sets our approach apart from traditional RL methodologies.
>
> Thank you for expressing your concern regarding the contribution of our paper. We would like to highlight our contribution and ask you for additional references to work that has done similar.
>
> To the best of our knowledge, our paper is the first to connect the existing work on Markov Convergence Theorem to the goal of learning long-term fair policies. Specifically we propose to impose necessary convergence criteria from the Markov Convergence Theorem and find a long-term fair policy by solving a constrained OP prior to deployment of the policy.
>
> To the best of our knowledge, our paper is the first to provide a comprehensive theoretically founded approach of how to find a long-term fair policy for the situation when dynamics are known or can be estimated with the following properties:
> * We can fix a single policy
> * By solving a constrained OP
> * That is guaranteed to converge to the stationary distribution
> * From any initial starting distribution
>
> We are happy to provide you with an overview of the technical challenges behind this. Our framework can be thought of as a three-step process. First, we show how to the characteristics of existing optimziation and fairness criteria as a function of the a stationary distribution. The second step involves transforming the definition of the fair characteristics into an optimization problem (OP) that allows to find a policy $\pi$ that induces a stationary distribution $\mu$, which adheres to the previously defined fairness targets. In the search of $\pi$, we showed that we first need to compute group-dependent kernel $T_{\pi}^s$, which, if the state space is finite, is a linear combination of assumed / estimated dynamics and distributions and policy $\pi$. We then compute the group-dependent stationary distribution $\mu_{\pi}^s$ via eigendecomposition. The third step involves solving the OP. Given the nature of our linear and constraint-based optimization problem, we can employ any efficient black box optimization methods for this class of problems.
>
> Our approach offers also several conceptual advantages to long-term fairness and trustworthiness: i) By establishing a time-invariant policy that reliably converges to a stationary distribution, we are offering a consistent decision-making framework that enhances stakeholder understanding and anticipation; ii) The model's versatility is demonstrated in its ability to accommodate various established fairness criteria, as it formulates long-term objectives and constraints from prior work as properties of a stationary distribution, providing a flexible approach to addressing fairness concerns; iii) The policy exhibits robustness to covariate shifts, making it effective across different distributions of $\mathbb{P}(X)$ and maintaining stability despite variations between feature distributions of different datasets.
>
> Regarding your concerns about the novelty of our work, we would like to ask you to kindly point us to related work that provides a similar contribution.
>
> We hope that our responses have adequately addressed your questions. We appreciate you raising your score, if our response has alleviated your concerns. Kindly let us know if there is anything else we can clarify.
>
> Best regards

---

### Meta-Review · Area_Chair_wnQy · 2023-12-02

**Metareview:**

This paper addresses the problem of designing long-term group fair policies beyond standard static fair policies.  The author consider the setting in which the decisions of ML algorithms can affect the individuals with data and therefore creates a dynamic between ML predictions and datasets. Leveraging the Markov chain formulation, and assuming the dynamic is learned, the authors formulate the design of long-term policies as an optimization problem and examine the approach with simulations.

Overall, all reviewers concur that the work addresses an important and relevant problem. The proposed framework is general and flexible, and the paper is well-written. On the other hand, there are varying levels of concern regarding the contributions. The technical contributions appear relatively incremental, especially when assuming that the dynamics can be estimated. A potentially more significant contribution lies in the modeling. However, as highlighted by reviewer ftJq, the assumptions and modeling choices of the paper necessitate further deliberation and justifications to strengthen the contributions.

**Justification For Why Not Higher Score:**

The contributions of the paper don't seem significant enough, at least based on the current presentation.

**Justification For Why Not Lower Score:**

N/A

---

### Decision · Program_Chairs · 2024-01-16

Reject